# Plant-Derived Nanovesicles from Soaked Rice Water: A Novel and Sustainable Platform for the Delivery of Natural Anti-Oxidant γ-Oryzanol

**DOI:** 10.3390/antiox14060717

**Published:** 2025-06-12

**Authors:** Jahnavi Ravilla, Soundaram Rajendran, Vidya M. Basavaraj, Greeshma Satheeshan, Janakiraman Narayanan, Thejaswini Venkatesh, Gopinath M. Sundaram

**Affiliations:** 1Department of Molecular Nutrition, CSIR-Central Food Technological Research Institute, Mysuru 570020, Karnataka, India; jahnavi1223@gmail.com (J.R.); soundaram103@gmail.com (S.R.); vidyashettymb19@gmail.com (V.M.B.); 2Department of Biochemistry and Molecular Biology, Central University of Kerala, Kasaragod 671320, Kerala, India; greeshmabbm072201@cukerala.ac.in (G.S.); thejaswinivenkatesh@cukerala.ac.in (T.V.); 3Department of Nanobiotechnology, Vision Research Foundation, No.18/41, Nungambakkam, Chennai 600006, Tamil Nadu, India; drnj@snmail.org; 4Academy of Scientific and Innovative Research (AcSIR), Ghaziabad 201002, Uttar Pradesh, India

**Keywords:** plant-derived exosome-like nanovesicles, rice water, gamma oryzanol, anti-oxidant activity

## Abstract

Gamma oryzanol (GO) is a natural anti-oxidant found in rice bran with potential health benefits. Conventional isolation of GO from rice bran requires the use of non-eco-friendly solvents such as acetone, ethyl acetate and hexane due to its low aqueous solubility. Further, nanoencapsulation of GO is required for the enhancement of stability and bioavailability. Plant-derived nanovesicles (PDNVs) are natural/intrinsic exosome-mimetic vesicles isolated from edible plants using green methods. Washed/soaked rice water (SRW) is often discarded as waste prior to cooking rice. However, traditional knowledge indicates its health-promoting anti-oxidant benefit, probably contributed by the presence of GO. Herein, for the first time, we isolated PDNVs from SRW by the cost-effective Polyethylene glycol 6000(PEG) precipitation method and demonstrated the presence of GO in PDNVs. In our initial screen, PDNVs were isolated from both rice grains (RGs) as well as the SRW of four different rice varieties, in which we identified the copious presence of GO in black RGs and brown SRW PDNVs. Both RG and SRW PDNVs were non-toxic to keratinocytes. SRW PDNVs displayed distinct cellular uptake mechanisms compared to RG PDNVs in human keratinocytes. Compared to native GO, brown SRW PDNVs containing GO displayed superior anti-oxidant activity in HaCaT keratinocytes, likely due to its enhanced cellular uptake. Overall, we describe here a waste-to-wealth green approach using an economical PEG method for the extraction of GO in bioavailable form. Given that oxidative stress is a driving factor for inflammation and related diseases, SRW PDNVs provide an affordable natural formulation for the treatment of diseases with underlying oxidative stress and inflammation.

## 1. Introduction

Gamma oryzanol (GO) is a natural anti-oxidant predominantly found in rice bran, with multiple other health benefits. GO is often isolated from rice bran oil as the raw material. GO is a mixture of ferulic acid esters and phytosterols in which cycloartenyl ferulates, campersteryl ferulates, 24-methylene cycloartanyl ferulate and beta-sitosteryl ferulate are the major constituents [1]. GO is widely recognized for its pharmacological properties, including anti-oxidant, anti-inflammatory, pro-wound healing and cholesterol-lowering effects [2,3,4]. Despite these promising benefits, the therapeutic application of GO is significantly limited by its poor oral bioavailability, primarily due to its hydrophobic nature, leading to poor aqueous solubility and limited intestinal absorption following oral administration [5]. In clinical and pre-clinical trials conducted in both human cohorts and animal models, approximately 80% of the administered GO is excreted in feces, highlighting its minimal absorption and poor bioavailability [6,7]. Notably, GO also undergoes extensive enzymatic hydrolysis in the gastrointestinal tract, resulting in its degradation into sub-compounds such as ferulic acid. Hence, despite the oral administration of high doses of GO, a very limited amount is detected in systemic circulation. In addition to this, GO is also subject to first-pass metabolism in the liver, leading to its transformation into secondary metabolites, further reducing its systemic bioavailability [8].

To overcome these limitations, various formulation strategies have been developed to increase the bioavailability of GO. These include nanoencapsulation, liposomal delivery, solid dispersions and microemulsion systems which are primarily aimed at enhancing its solubility, preventing premature degradation and enhancing the bioavailability [9,10,11,12]. Nevertheless, these methods also suffer from demerits such as high production costs, stability concerns, short shelf life and the use of surfactants which may be toxic or an irritant in long-term use [13]. Apart from bioavailability issues, the extraction process employed for GO also poses its own set of challenges. GO is conventionally isolated from rice bran/bran oil via solvents that are not eco-friendly [14]. Firstly, crude rice bran oil is extracted from rice bran using hexane, from which GO is isolated via a two-step crystallization process which involves the use of acetone and hexane again [15]. Such solvent systems pose significant health risks and have a considerable environmental foot print [16]. Recently, supercritical fluid extraction (SFE) was compared to the conventional extraction procedure for isolating GO from rice bran. Compared to conventional hexane extraction, SFE produced higher GO content with a stronger anti-oxidant effect and greater inhibition of cancer cell growth [17]. However, SFE also suffers from high equipment cost, operational complexity and scale-up challenges, necessitating further research into innovative methods for the eco-friendly extraction of GO in bioavailable form [18].

Plant-derived nanovesicles (PDNVs) are exosome-mimetic vesicles isolated from edible plants with a size range between 100 and 500 nm [19,20]. PDNVs are garnering increasing interest in plant bioactive research due to their ability to deliver plant bioactives in bioavailable form, while also enabling the extraction of these bioactives without the use of non-ecofriendly solvents [21]. Notably, PDNVs isolated from ginger, broccoli, grapefruit and lemon have been reported to naturally contain their respective bioactives including 6-gingerol/6-shogaol, sulforaphane, naringin/naringenin and ascorbic acid, respectively, which are well known for their anti-oxidant, anti-inflammatory and anti-cancer properties [19,22,23,24,25,26]. In particular, PDNVs are conventionally isolated through a multi-step process that begins with the preparation of an aqueous homogenate of the plant material, followed by differential centrifugation at forces ranging from 2000× *g* to 10,000× *g* to eliminate plant fibers, cellular debris and micro vesicles. The resulting supernatant is then subjected to ultracentrifugation to pellet the PDNVs. For high purity, an additional sucrose gradient centrifugation step may be employed [27,28]. While this method is environmentally friendly, its scalability and cost-effectiveness remain a major concern. Due to this, we recently developed a polymer (PEG6000) precipitation protocol for PDNV isolation, which obviates the need for ultracentrifugation [27,29].

Soaked rice water (SRW), a by-product generated during the soaking of rice, is often regarded as a form of industrial and household waste. Traditional knowledge and limited scientific evidence suggest that SRW possesses several health-promoting properties, including anti-oxidant, anti-inflammatory, photoprotective, and anti-aging effects, particularly in the context of skin health [30,31]. These benefits have been attributed to the presence of amino acids, minerals and phenolic compounds in SRW. Although GO is rich in rice bran and bran oil, its presence in SRW is considered unlikely due to its low aqueous solubility. In this study, we report for the first time the selective isolation of PDNVs from SRW using a cost-effective PEG-based precipitation method [27]. Out of four rice varieties used for SRW PDNV isolation, brown rice SRW-derived PDNVs were found to contain high GO content and exhibited functional activity by ameliorating hydrogen peroxide (H_2_O_2_)-induced oxidative stress in HaCaT keratinocytes. For comparative purposes, PDNVs were also isolated from the intact rice grains (RGs) of all four varieties. Interestingly, SRW PDNVs displayed superior characteristics compared to RG PDNVs, including more rapid cellular internalization and distinct uptake mechanisms and differential GO content. These findings suggest that SRW PDNVs may represent authentic extracellular vesicles/exosomes that diffuse from rice grains during soaking. On the other hand, RG PDNVs likely comprise a heterogenous population of both native and de novo nanovesicles formed due to cellular disruption during homogenization, consistent with previous observations [32].

## 2. Materials and Methods

### 2.1. Materials

White rice (Pusa Basmati 1509), Black rice (Karuppu Kavani), Red rice (UMA M016) and Brown rice kernels were procured from Kerala, India. Gamma oryzanol (#O0172) standard was procured from TCI chemicals (Tokyo, Japan). PEG 6000 was purchased from Sigma Aldrich (St. Louis, MO, USA) (#81260). Dil C18 stain (1,1’-Dioctadecyl-3,3,3’,3’-Tetra-methylindo-carbocyaninie Perchlorate) was from Thermofisher Scientific (Waltham, MA, USA) (#D3911). HaCaT keratinocytes were a kind gift from Prof. Colin Jamora, InStem, Bangalore, India. All HPLC-grade solvents were from Merck Millipore (Burlington, MA, USA). Milli-Q grade water was used in all the experiments.

### 2.2. PDNV Isolation from Rice Grains (RGs) and Soaked Rice Water (SRW)

PDNVs were isolated from white (Pusa Basmati 1509), black (Karuppu Kavuni), red (UMA M016) and brown rice kernels procured from Kerala, India. Rice grains were de-husked using a small-scale mill to preserve the bran layer. For PDNV isolation, intact rice kernels were washed and soaked in twice the volume of double distilled water for 12 h at room temperature. The soaked rice water (SRW) was harvested by filtration with a muslin cloth. While the left-over soaked rice grains were used for PDNV isolation from intact grains, SRW was used for the extraction of actively secreted PDNVs. Purification of PDNVs from intact rice grains was carried out as described earlier [27]. In brief, soaked rice grains were homogenized in a blender, followed by filtration through a nylon mesh (100 μm pore size). Filtered rice grain homogenate and SRW were subjected to sequential centrifugation at 2000× *g* for 10 min, 6000× *g* for 20 min and 10,000× *g* for 30 min to remove cellular debris, apoptotic bodies and macrovesicles, leading to the preparation of a clarified S10 extract. PDNVs in S10 extract were precipitated by the addition of polyethylene glycol 6000 (PEG6000) to a final concentration of 10% (*w*/*v*) [27,29]. Samples were allowed to be incubated at 4 °C overnight with gentle shaking. PDNVs were sedimented by centrifugation at 10,000× *g* for 30 min at 4 °C. The pellet containing PDNVs was suspended in distilled water, and stored at 4 °C, up to a week. For long-term storage, PDNV suspensions were mixed with a cryoprotectant, Trehalose (final concentration, 50 mM), lyophilized and stored at −80 °C as described earlier [33]. To track the intracellular uptake, PDNVs were labeled with a lipophilic dye, Dil, by addition of Dil to S10 extract, to a final concentration of 0.1 μM before PEG precipitation. PDNVs quantification was carried out by measuring the total lipid content with Sulpho-phospho-vanillin assay [34].

### 2.3. PDNV Characterization for Size, Zetapotential and Morphology

Nanovesicles’ size distribution, polydispersity index (PDI) and overall surface charge potential (zeta potential) were assessed using a Malvern zeta sizer nano ZS (Malvern Instruments, Malvern, UK). PDNV samples were diluted 100-fold in Milli-Q water, and triplicate measurements were taken at room temperature for each sample. Particle sizes and zeta potentials of at least three independent batch purifications were measured, and their mean ± standard deviation was calculated. SEM analysis of PDNVs was carried out as described earlier [35]. In brief, isolated PDNVs were reconstituted in 1% tannic acid (Sigma #403040) and 2.5% glutaraldehyde (Sigma #G6257) in 100 mM sodium cacodylate buffer (Sigma #C0250) (pH-7.2) and fixed for 30 min. Fixed and diluted samples were spotted onto Nucleopore track-etched membrane (Avanti lipids #610005) (100 nm pore size) and air-dried overnight. PDNV-coated membranes were placed on an SEM stub which was sputter-coated with a gold layer (5 nm) using an MCM 100 sputter coater (SEC). PDNV images were acquired using an advanced Scanning Electron Microscopy (SEC Co. Ltd., SNE 3200M, Suwon, Republic of Korea) instrument.

### 2.4. Total Lipid and Total Protein Determination of PDNVs

Quantification of total lipid content in PDNVs was performed by colorimetric reaction of sulphuric acid and phospho-vanillin with lipids as previously reported [36]. Olive oil in absolute ethanol (10 µg/µL) served as a lipid standard solution. Lipid contents of isolated PDNVs were extracted using chloroform and methanol (1:2, *v*/*v*) and the extracts were completely air-dried. An equal volume of ethanol was added to all standards and samples followed by the addition of 200 µL of 98% sulphuric acid and incubated at 95 °C for 15 min, and immediately cooled at −20 °C for 15 min. A 500 µL of phospho-vanillin reagent (150 mg of vanillin (Sigma #V1104, St. Louis, MO, USA) dissolved in 20 mL of ethanol and mixed with 85% ortho-phosphoric acid) was added into each tube, vortexed and incubated for 30 min in the dark at room temperature. The optical density was read at 540 nm and the amount of lipids was determined in µg/mL lipid equivalence.

The total soluble protein content was measured by copper-based bicinchoninic acid assay [37], using a BCA assay kit (G Bioscience, St. Louis, MO, USA). To extract the proteins, PDNVs were treated with RIPA cell lysis buffer (4× containing 100 mM Tris pH 7.6, 2 M NaCl, 20 mM EDTA, 4% Triton X100, 4% sodium deoxycholate and 0.4% SDS for 20 min at room temperature. Samples were centrifuged at 10,000× *g* for 15 min at 4 °C to remove the insoluble contents and the supernatant containing extracted protein was collected. Total protein estimation of the extract was performed according to the BCA kit manufacturer’s protocol, the absorbance was read at 562 nm and the concentration was determined by a standard curve obtained using fixed concentrations of bovine serum albumin (BSA).

### 2.5. Thin-Layer Chromatography Analysis of PDNV Lipids

The thin-layer chromatography (TLC) was carried out as described earlier [38,39]. A chloroform and methanol (1:2) mixture was added to PDNVs of 1 mg lipid equivalent and vortexed thoroughly. The mixture was subjected to centrifugation at 5000× *g* for 10 min at room temperature, from which the organic phase containing the extracted lipids was collected, dried and reconstituted in chloroform. The lipid extracts were spotted on silica gel F_254_-TLC plates (Merck, Rahway, NJ, USA, #1.05554) and resolved using a mobile phase consisting of chloroform/methanol/acetic acid (95:4.5:0.5, *v*/*v*). Plates were dried and sprayed with a solution containing 10% copper sulphate and 8% phosphoric acid. Lipid bands were visualized by charring the plates at 120 °C for 10 min.

### 2.6. HPLC Analysis of γ-Oryzanol from PDNVs

The extraction of γ-Oryzanol from PDNVs was performed following the method described in a previous study with slight modification [40]. Briefly, about 10 mg lipid equivalent PDNVs were mixed with ethyl acetate at a volume ratio of 1:4 (PDNVs/ethyl acetate, *v*/*v*), followed by 30 s extensive vertexing at room temperature. The mixture was subjected to ultrasonication (Qsonica Q800R3 sonicator, Fisher Scientific, Pittsburg, PA, USA) for 10 min at 50 °C with a pulsed operation setup (10 s ON and 10 s OFF) and centrifuged at 10,000× *g* for 10 min. The resulting organic phase was collected, evaporated using a speed-vac concentrator and further used for HPLC analysis.

The γ-Oryzanol from PDNVs was analyzed using Shimadzu chromatography (Kyoto, Japan) equipped with reverse-phase HPLC-UV-C18 column (Shimadzu, Shim-pack Solar, 5 µm C18, 4.6 × 250 mm, HSS), resolved with an isocratic elution of methanol/acetonitrile/dichloromethane/acetic acid (50/45/3/3, *v*/*v*/*v*/*v*), at a flow rate of 1 mL/min by an LC-20AD pump cabinet, at a column temperature of 40 °C (CTO-20AC); the UV detector was set at 325 nm (SPD-M20A), and a computer software application (LC solutions version 5.106) was used [41]. The standard solutions of 0.0625, 0.125, 0.25, 0.5 and 1 µg were analyzed under the same HPLC conditions. An aliquot of 10 µL for each sample reconstituted in ethyl acetate (100 µL) was applied to the HPLC column. The standard chromatogram/calibration curve, which was based on the peak area of the γ-Oryzanol standards, was used to estimate the amount of γ-Oryzanol in the PDNV extracts. All extractions and HPLC analysis were performed at least in triplicate, and the data were calculated as mean ± standard deviation.

### 2.7. Cell Culture

HaCaT cells were cultured in Dulbecco’s Modified Eagle Medium (DMEM) (Himedia, Maharashtra, India, #AL007S) containing 10% fetal bovine serum (FBS, Gibco #A5256701, Waltham, MA, USA) and 1% Penicillin-Streptomycin (PS, Himedia #A014, Thane, India). Cells were maintained in a 5% CO_2_ incubator at 37 °C and passaged at least twice a week.

### 2.8. Cell Viability Assay

The effect of PDNVs on HaCaT keratinocyte proliferation was assessed using MTT (3-[4,5-dimethylthiazol-2-yl]-2,5-diphenyltetrazolium bromide) (Invitrogen, Thermo Fisher Scientific, Waltham, MA, USA, #M6494) cell viability assay. HaCaT keratinocytes (1 × 10^4^ cells/well) were seeded in 96-well plates and incubated overnight for effective adhesion. The cells were treated with only media (mock) or with increasing concentrations of RG and SRW PDNVs for 24 h. MTT reagent was added to each well (10 μg/well) and cells were further incubated for 1 h. The resulting insoluble formazan crystals were dissolved in DMSO, and absorbance was measured at 570 nm.

### 2.9. Intracellular Uptake Assay and Inhibitor Studies

The internalization of PDNVs by HaCaT cells was confirmed by tracking the Dil-labeled PDNVs. HaCaT cells were seeded in 24-well tissue culture plate at a uniform density of 5 × 10^4^ cells per well and incubated at 37 °C in a CO_2_ incubator to attain 75–80% confluency. Then Dil-labeled PDNVs (160 µg lipid equivalent) were added to the wells and incubated for different time points (0, 0.5, 1, 2, 4 and 8 h). After incubation for defined time points, the spent media was removed, and the cells were washed gently with PBS to remove the unbound Dil-PDNVs, and fixed with 4% paraformaldehyde for 10 min. The 0 h time point served as a negative control, wherein cells treated with PDNVs were immediately washed to remove unbound PDNVs, and processed for fixation. The cells were counter-stained with DAPI (Sigma Aldrich, St. Louis, MO, USA) and were imaged using a fluorescence microscope (Olympus IX73 Olympus corporation, Tokyo, Japan) under 60× magnification with a TRITC filter. For endocytosis inhibitor experiments, cells were treated with specific endocytosis inhibitors before PDNVs treatment to analyze the effect of inhibitors in intracellular entry of PDNVs. Treatment with PDNV alone without inhibitors served as mock. The concentration and incubation period of inhibitors used were Methyl-β-cyclodextrin (MβCD) (Sigma Aldrich #C4555, St. Louis, MO, USA) at 10 μM (15 min), Amiloride (AML) (Sigma Aldrich #A7410, St. Louis, MO, USA) at 375 μM (1 h), Chlorpromazine (CPZ) (Sigma Aldrich #C8138, St. Louis, MO, USA) at 22.5 μM (15 min) and Indomethacin (IND) (Sigma Aldrich #I7378, St. Louis, MO, USA) at 225 μM (1 h). Cells were washed and treated with PDNVs (concentration 160 μg/mL) for 4 h. The post-treatment procedures were followed as mentioned above.

### 2.10. Measurement of Intracellular ROS

To assess the anti-oxidant activity of PDNVs, HaCaT cells were incubated with or without RG/SRW PDNV (480 μg/mL) or γ-Oryzanol (300 ng/mL) alone for 12 h. Oxidative stress was then induced by treatment with 500 μM of hydrogen peroxide (H_2_O_2_) for 3 h. Cells were subsequently incubated with 20 μm DCFH2-DA (2′,7′-Dichlorodihydrofluorescein diacetate) (Sigma #D6883, St. Louis, MO, USA) for 20 min and cells were gently rinsed with PBS. Fluorescence images were acquired using an inverted fluorescence microscope (Olympus IX73, Olympus corporation, Tokyo, Japan) using FITC channel at 20× magnification. The total green fluorescence intensity in multiple random images taken was quantified using ImageJ software (Version 1.54g, NIH, Bethesda, MD, USA).

### 2.11. Statistical Methods

The data shown here are the average result of three or more independent experiments with minimum technical triplicate measurements performed in each assay. Data are expressed as means ± standard deviation (SD) and the statistical analysis was performed with the ANOVA algorithm in GraphPad Prism8 software.

## 3. Results and Discussion

### 3.1. Isolation of PDNVs from Rice Grain (RG) and Soaked Rice Water (SRW) from Different Rice Varieties

The hypothesis that SRW may contain actively secreted PDNVs stemmed from a recent finding which demonstrated the presence of the multivesicular bodies containing exosome-like structures (PDNVs) within the aleurone layer of rice seeds by electron microscopy [42]. Considering that exosomes are communication vehicles across different cell types, we anticipated the spontaneous release of these PDNVs in SRW when rice grains were soaked in water with a purpose to turn waste (SRW) into a wealthy resource in the form of PDNVs. Hence, de-husked rice grains with intact bran layer were soaked in distilled water for 12 h. SRW was harvested by filtering out the soaked rice grains through a muslin cloth, and PDNVs were purified from the SRW by the differential centrifugation-coupled PEG precipitation method described earlier [27] (Figure 1A). In this study, four different rice varieties that are commonly used for consumption, such as white, red, brown and black rice, were used for PDNV isolation. In addition to SRW PDNVs, we also purified PDNVs from the left-over soaked rice grains (RGs) for a comparative assessment of the relative levels of PDNVs released into SRW, versus PDNVs present in the left-over soaked rice grains (Figure 1A) [43]. The purified PDNVs were first assessed for their average size and stability (zeta potential values) via differential light scattering analysis. As shown in Figure 1B,D, both RGs and SRW contained PDNVs with a size range of ~ 110 to 317 nm obtained from three independent batches of rice varieties. PDNVs isolated from brown RG and SRW and red RG displayed the smallest size average (<150 nm), whereas white RG/SRW, red SRW and black RG/SRW PDNVs displayed a size average >250 nm (Figure 1B,D). The zeta potential values of brown RG and SRW PDNVs and black RG PDNVs were much lower than all the other PDNVs, which ranged between ~ −4.0 mV and ~ −7.0 mV. All other SRW PDNVs (white, red and black) showed higher zeta potential values between −19.0 mV and −28.0 mV, suggesting their increased colloidal stability (Figure 1C,D) [44]. The surface morphology of all PDNVs was resolved through a scanning electron microscope, in which PDNVs’ spherical structures with smooth surface morphology embedded onto the membrane (Figure 1E). The differences in PDNV characteristics (size and zeta potential) between RG and SRW PDNVs could be attributed to their distinct origin of biogenesis. In plants, authentic exosomes/extracellular vesicles (EVs) are known to be secreted within the apoplastic fraction. These EVs are often purified form the apoplastic fraction of the plant material by differential ultracentrifugation. Notably, the apoplastic fractions of plants are collected via gentle by vacuum infiltration. On the other hand, purification of PDNVs requires an initial homogenization step of the plant material under harsh conditions, which has also been proposed to allow the formation of de novo nano-sized vesicles (PDNVs) resulting from the “re-mixing” of disrupted cellular membranes during homogenization [32]. Hence, SRW PDNVs could possibly represent authentic exosomes/EVs released from the aleurone layer while soaking, whereas the RG PDNVs could likely represent a mixture of authentic exosomes/EVs along with PDNVs formed de novo during the homogenization.

PDNVs are known to contain lipids, secondary metabolites, protein and small RNA fractions (microRNAs) which are known to execute its effect on cells. Hence, we profiled the other key bioactive ingredients of isolated PDNVs such as lipids, proteins and microRNAs. Notably, the measurement of lipid content has been exploited as a surrogate method to quantify the relative yield of exosomes as well as PDNVs [36]. Hence, we measured the total lipid content of PDNVs, and higher lipid content was observed in RG PDNVs compared to SRW PDNVs (Figure 2A), suggesting higher PDNV yields. Notably, black RG PDNVs displayed the highest lipid content compared to the other PDNVs. The overall relative yield of PDNVs was much lower in all SRW PDNVs without a significant difference between the rice varieties used (Figure 2A). The relative composition of lipids in PDNVs was further profiled through TLC analysis (Figure 2B). We noted the relative lipid profiles for SRW PDNVs were distinct from RG PDNVs, further substantiating the distinct origin of PDNVs. The protein contents of PDNVs were quantified by BCA assay as described in the Materials and Methods section. The overall protein content of RG PDNVs was significantly higher compared to SRW PDNVs (Figure 2C). The protein to lipid (PL) ratio has been employed as a tool to distinguish authentic mammalian exosomes from other vesicles of non-exosome origin, such as apoptotic bodies and microvesicles, in which lesser the PL ratio is characteristic of exosomes/extracellular vesicles [45,46]. In our analysis, we noted a significantly lesser PL ratio for SRW PDNVs compared to RG PDNVs (Figure 2D). Taken together, all the above results indicate that SRW PDNVs may represent authentic exosomes/EVs released by rice grains during soaking. We could not recover significant small RNA fractions from SRW PDNVs and they were therefore excluded from further characterization.

### 3.2. Both RG and SRW PDNVs Contain GO

SRW is well known for its health benefits, especially for its anti-aging, anti-oxidant and anti-inflammatory activities in skin, and these activities are proposed to be due to its GO content [1,31]. Due to its low aqueous solubility, it is unlikely that GO exists in its native form in SRWs. Considering several key plant bioactives which exhibit poor aqueous solubility have been found to be present in soluble form within PDNVs, we hypothesized that GO must be present within RG and SRW PDNVs to exist in a soluble form in an aqueous environment [19,47,48]. To test this, both RG and SRW PDNVs were investigated for their GO content. GO from both PDNVs was selectively extracted with ethyl acetate and subjected to HPLC analysis as described in the Materials and Methods section. GO is a mixture of four major individual isomers, namely (a) cycloartenylferulate, (b) 24-methylenecycloartanyl ferulate, (c) campsterylferulate and (d) β-sitosteryl-ferulate [49]. HPLC analysis of standard GO revealed four separable peaks at specific retention times (T_R_) corresponding to each isomer, namely peak (a) cycloartenylferulate at 17.9 min, peak (b) 24-methylenecyclo-artanyl ferulate at 19.3 min, peak (c) campesteryl ferulate at 19.9 min and peak (d) β-sitosteryl-ferulate at 22.3 min, with a dose-dependent increase in peak intensity with increasing concentration of GO (Appendix A). Notably, both RG and SRW PDNVs displayed the presence of four peaks corresponding to all isomers of GO at their specific T_R_, suggesting the presence of GO within PDNVs (Figure 3A(a–d)). When peak areas from multiple batches of RG and SRW PDNVs were quantified, we noted that black RG PDNVs possessed the highest concentration (~70–300 ng GO isomers/mg of lipid equivalent) of GO followed by brown SRW PDNVs (~30–100 ng GO isomers/mg of lipid equivalent) compared to all other PDNVs (Figure 3B(a–d),C). Notably, we observed significantly higher campesteryl ferulate content in brown SRW PDNVs compared to other SRW PDNVs (Figure 3B(c)). The increased GO content in black RG and brown SRW PDNVs could likely be due to the higher GO content in black and brown rice compared to other rice varieties employed here [50,51,52]. Further, supplementation of brown rice bran powder in the standard diet has been shown to improve the overall anti-oxidant status and reduce the obesity of individuals in a double blind randomized clinical trial, substantiating our results [53]. Next, we assessed the relative distribution of GO isomers within each PDNV to determine if PDNVs isolated from different rice varieties exhibit variation in isomer composition. The relative proportion of the four GO isomers remained largely similar between RG and SRW PDNVs from red and black rice varieties (Figure 3D). However, white rice SRW PDNVs showed higher cycloartenyl ferulate and lower campesteryl ferulate levels compared to RG PDNVs, while brown SRW PDNVs exhibited the opposite trend (Figure 3D). These findings indicate that SRW PDNVs from white and brown rice possess distinct GO isomer profiles, likely reflecting their different origins. This also further strengthens our point that SRW PDNVs may represent authentic extracellular vesicles (EVs) released while soaking, whereas RG PDNVs could comprise a heterogenous mix of EVs as well as nanovesicles generated de novo during homogenization [32]. However, the authentic extracellular vesicular nature of SRW PDNVs may further need to be confirmed by investigating the emerging protein markers that are selectively known to be present in plant exosomes, such as TET8, syntaxin PEN1 and the ABC transporter PEN3 [54]. Based on the above results, SRW can be considered a valuable source for isolating PDNVs enriched with soluble GO isomers.

### 3.3. Both RG and SRW PDNVs Are Non-Toxic to Keratinocytes and Are Taken Up by Keratinocytes

Amongst its multiple health benefits, GO has garnered significant attention for its anti-oxidant and anti-aging properties, particular in promoting skin health—making it a popular natural ingredient in cosmetic formulations [55]. Hence, to test the anti-oxidant activity of PDNV-associated GO on skin, we used HaCaT keratinocytes, a spontaneously immortalized human keratinocyte cell line widely used for studies in skin biology as a model system [56]. Given that PDNVs are typically derived from edible plants, they demonstrate excellent biocompatibility and low toxicity in both in vitro and in vivo models [43]. Hence, we first investigated the effect of RGs and PDNVs on the cell viability of human keratinocytes under basal conditions. Since black RG PDNVs and brown SRW PDNVs contained high levels of GO, we chose these PDNVs for further studies. PDNVs were also purified from black SRW and brown RGs as additional controls in subsequent assays. HaCaT cells were incubated with different concentrations (lipid equivalent) of indicated PDNVs for 24 h and cell viability was assessed by MTT assay. In line with previous studies, brown RG/SRW PDNVs and black RG PDNVs showed no detectable toxicity up to 200 μg/mL in cell viability assays with HaCaT cells, while a mild but significant decrease in cell proliferation was observed with black SRW PDNVs. There was a dose-dependent decrease in cell viability with brown RG PDNVs at higher concentrations, likely due to their distinct phytochemical/lipid/protein composition. Since a dose-dependent decrease in cell viability was not observed in black SRW PDNVs, it is possible that they induce keratinocyte differentiation-coupled cell cycle arrest without causing overt cytotoxicity (Figure 4) [57].

Intracellular delivery of GO is an essential step to achieve its anti-oxidant potential. Hence, we next investigated if the RG/SRW PDNVs can spontaneously be taken up by HaCaT cells. To track the intracellular uptake of these vesicles, PDNVs were labeled with a fluorescent dye, Dil (1,1′-Dioctadecyl-3,3,3′,3′-Tetramethylindocarbocyanine Perchlorate), which exhibits very weak fluorescence under native conditions but exhibits strong fluorescence once it is incorporated into membrane structures. Hence, the presence of positive Dil stain on cells indicates successful cellular uptake of PDNVs. When HaCaT cells were incubated with Dil-labeled PDNVs, successful cellular uptake of SRW PDNVs from brown and black rice was detected as early as 30 min, and this cellular uptake reached saturating intensities within 8 h of incubation with PDNVs (Figure 5A). On the other hand, black and brown RG PDNVs displayed comparatively slower uptake kinetics but reached saturating uptake levels at 8 h (Figure 5A). The intracellular uptake kinetics is in agreement with prior studies wherein PDNVs from edible plants were shown to enter cells within hours of incubation [43,58,59,60,61,62]. The relatively faster uptake behavior of SRW PDNVs compared to RG PDNVs may further support their authentic extracellular vesicular nature. Encouraged by these results, we next investigated the specific uptake mechanism of both PDNV types to determine whether the enhanced uptake of SRW PDNVs is attributable to a distinct internalization pathway. Hence, HaCaT cells were exposed to selective inhibitors of different endocytic pathways, such as methyl-beta-cyclodextrin (MβCD), Amiloride (AML), Chlorpromazine (CPZ) and Indomethacin (IND), which are known to inhibit cholesterol/lipid raft-mediated endocytosis, macropinocytosis, clathrin-mediated endocytosis and caveolin-mediated endocytosis, respectively [63]. After treatment with inhibitors, cells were incubated further with brown and black RG/SRW PDNVs and observed after 4 h of incubation. As predicted, we observed distinct uptake mechanisms between RG and SRW PDNVs (Figure 5B). We noted profound inhibition of both brown and black SRW PDNVs by MβCD but black and brown RG PDNVs were resistant to MβCD-mediated endocytic inhibition (Figure 5B). This indicates SRW PDNVs primarily utilize the cholesterol/lipid raft-mediated endocytic pathway for their uptake, which is akin to mammalian exosomes [64]. Notably, SRW PDNVs were also partly inhibited by CPZ and IND, suggesting SRW PDNVs may also utilize clathrin, caveolae-mediated endocytosis and macropinocytosis. On the other hand, black RG PDNVs were specifically inhibited by IND, whereas significant inhibition of brown RG PDNVs was observed with AML and CPZ, suggesting different uptake mechanisms utilized by brown and black RG PDNVs (Figure 5B). These results mirror the results obtained from other studies wherein distinct PDNVs utilize different pathways for their cellular uptake. For instance, apple PDNVs were shown to be exclusively dependent on clathrin-dependent endocytosis, whereas corn PDNVs were dependent on cholesterol/lipid raft-mediated endocytosis, and turmeric PDNVs utilize all the pathways mentioned above for their uptake [63,65,66]. Hence, it is possible that the cellular uptake of PDNVs could be determined by multiple factors, which include the intrinsic factors associated with PDNVs such as the specific lipid composition, size, surface proteins and charge, whereas other factors such as the target cell type, membrane composition and endocytic activity could profoundly affect their internalization [67].

### 3.4. Brown SRW PDNVs Prevent H_2_O_2_-Induced Oxidative Stress in HaCaT Keratinocytes Better than Native GO

H_2_O_2_ is a reactive oxygen species (ROS) commonly used to induce oxidative stress in cellular models. At elevated concentrations, H_2_O_2_ disrupts the redox homeostasis and generates hydroxyl free radicals, leading to oxidative stress. Given that GO is known for its anti-oxidant activity and several earlier reports have demonstrated the anti-oxidant activity of GO in H_2_O_2_-treated cells, we next investigated if brown SRW or black RG PDNVs containing relatively high levels of GO exert better anti-oxidant activity compared to native GO. A wide range of GO concentrations, from 5 μg/mL to 200 μg/mL, has been employed to demonstrate its anti-oxidant activity across various cell types [1,68,69]. In our cell culture-based anti-oxidant assay, HaCaT keratinocytes were treated with a low concentration of GO (0.3 μg/mL), chosen deliberately to elicit minimal anti-oxidant effects and to serve as a baseline for effective comparison. In parallel, cells were also treated with equal concentrations of PDNVs (480 μg/mL lipid equivalent) isolated from brown/black SRW or RGs. The actual GO content in black RG, black SRW, brown RG and brown SRW PDNVs, were approximately 0.305 μg/mL, 0.068 μg/mL, 0.04 μg/mL and 0.138 μg/mL, respectively (Figure 6B). Cellular ROS levels were quantified using the fluorescence intensity of the ROS-sensitive probe DCFH2-DA. As noted in Figure 6A, H_2_O_2_ treatment markedly increased intracellular ROS in HaCaT cells, while pretreatment with free GO led to a significant reduction in ROS levels, despite the suboptimal concentration of GO used. Interestingly, pretreatment with black RG and brown SRW PDNVs resulted in a pronounced decrease in ROS accumulation, whereas black SRW and brown RG PDNVs exhibited only mild anti-oxidant effects. These differences may be attributed to two key factors. First, the enhanced anti-oxidant activity of black RG PDNVs likely stems from their relatively high GO content (Figure 3B). Second, the potent anti-oxidant response elicited by brown SRW PDNVs, despite their lower GO content, could be due their superior cellular uptake kinetics and diverse endocytic routes (Figure 5), which may facilitate more efficient GO delivery into HaCaT cells. Lastly, it is also important to acknowledge that the anti-oxidant effects observed may not be solely attributable to the GO content. Other bioactives present within the PDNVs, such as miRNAs, lipids, proteins and phytochemicals, are also known to contribute to their anti-oxidant functionality [43,70,71]. Given that the bioactive cargo of SRW PDNVs is expected to differ significantly from that of RG PDNVs, these compositional differences could also be the underlying reason for the distinct anti-oxidant responses observed. Collectively, our finding suggests that brown SRW PDNVs may represent a natural, bioavailable, cost-effective, and sustainable “waste-to-wealth” source of functional bioactives including GO, with therapeutic relevance in oxidative stress-related skin conditions.

## 4. Conclusions

In this study, we present a novel, eco-friendly approach for isolating PDNVs from soaked rice water, a commonly discarded by-product of rice processing. We demonstrate, for the first time, that SRW PDNVs are enriched with bioactive GO isomers, and exhibit physiochemical properties and biological functions better than PDNVs derived from whole RG. Within the four rice varieties, brown rice SRW PDNVs not only encapsulate GO naturally but possess enhanced cellular uptake and anti-oxidant effects in human keratinocytes. These findings also establish SRW as a sustainable and cost-effective source of functional PDNVs, with significant potential for use in cosmeceutical and nutraceutical applications. Since GO exhibits anti-cancer, anti-oxidant, anti-aging and anti-inflammatory properties, this study could be further extended to evaluate the SRW PDNVs’ potential in relevant disease models such as skin inflammation, UV-induced skin aging/allergies, oxidative stress-related disorders and cancer. To evaluate the ability of SRW PDNVs to deliver GO in skin tissues to exert its anti-aging activity, an ex vivo porcine ear skin model can be used due its resemblance to human skin [27,29,33,34,35,36,37,38,39,40,41,61].

## Figures and Tables

**Figure 1 antioxidants-14-00717-f001:**
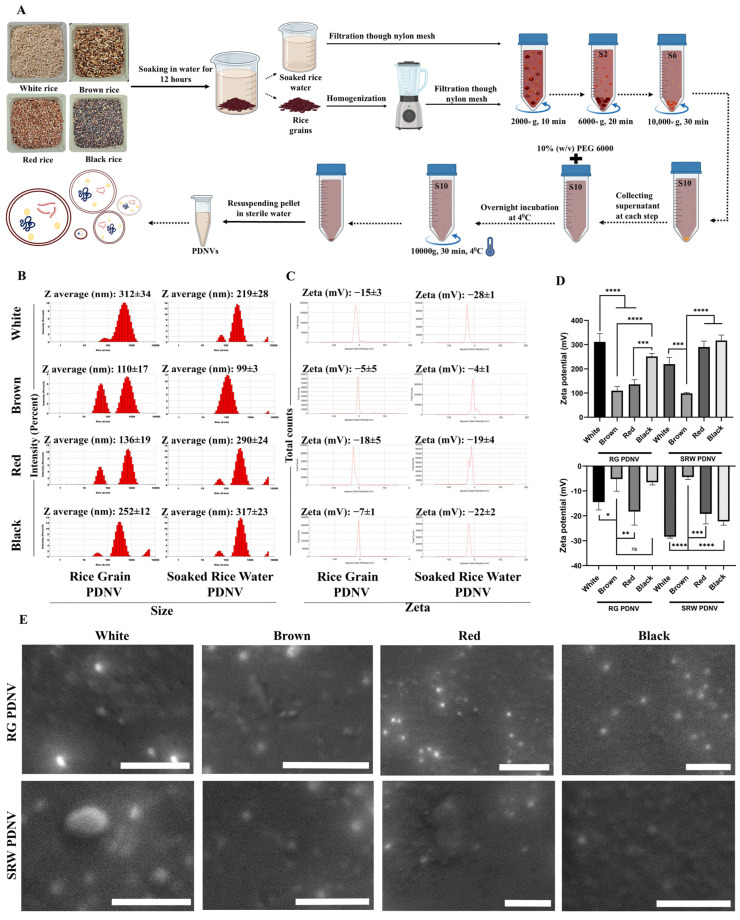
Isolation of PDNVs from rice grain (RG) and soaked rice water (SRW) from indicated rice varieties. (**A**) Schematic depicting the methodology used for the isolation of rice grain (RG) and soaked rice water (SRW) PDNVs (S2, S6, S10 refer to supernatant obtained after centrifugation at 2000× *g*, 6000× *g* and 10,000× *g*, respectively). (**B**) Representative image of the average size distribution of PDNVs isolated from RG and SRW. (**C**) Representative image of average zeta potential of PDNVs isolated from RG and SRW. (**D**) Size and zeta potential values from three independent batches of RG or SRW PDNVs. (**E**) Scanning electron microscopic images of the RG and SRW PDNVs. Scale bar—2 μm. n.s. means nonsignificant. * *p* < 0.05, ** *p* < 0.01 *** *p* < 0.001 and **** *p* < 0.0001.

**Figure 2 antioxidants-14-00717-f002:**
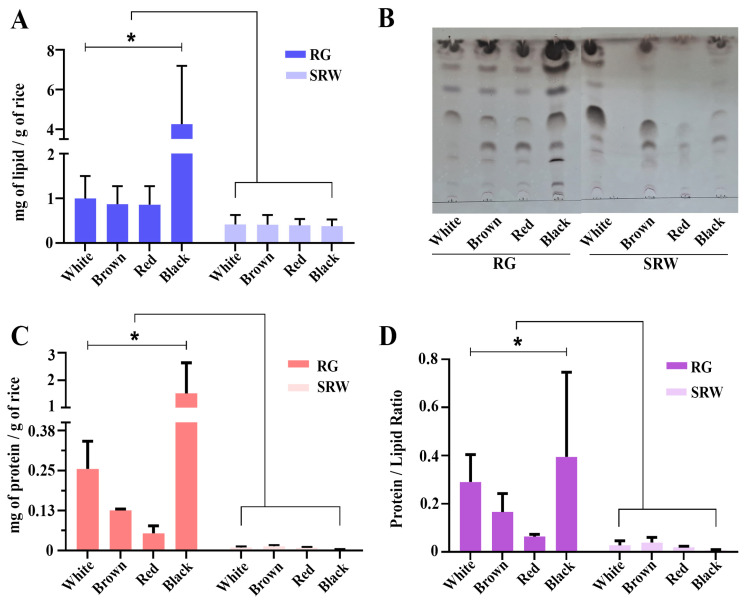
Biochemical characterization of RG and SRW PDNVs. (**A**) Total lipid content of isolated PDNVs assessed by phospho-vanillin method. (**B**) Total lipids were extracted from indicated PDNVs and resolved using a Silica Gel 60 F_254_ plate. (**C**) Total protein content of isolated PDNVs estimated by BCA method. (**D**) Protein to lipid ratio was calculated based on the concentrations of PDNVs obtained with BCA and phospho-vanillin assay methods. All figures represent data from an average of three independent PDNV batches isolated from different rice varieties. All figures represent data from three independent PDNV batches isolated from different rice varieties. * *p* < 0.05.

**Figure 3 antioxidants-14-00717-f003:**
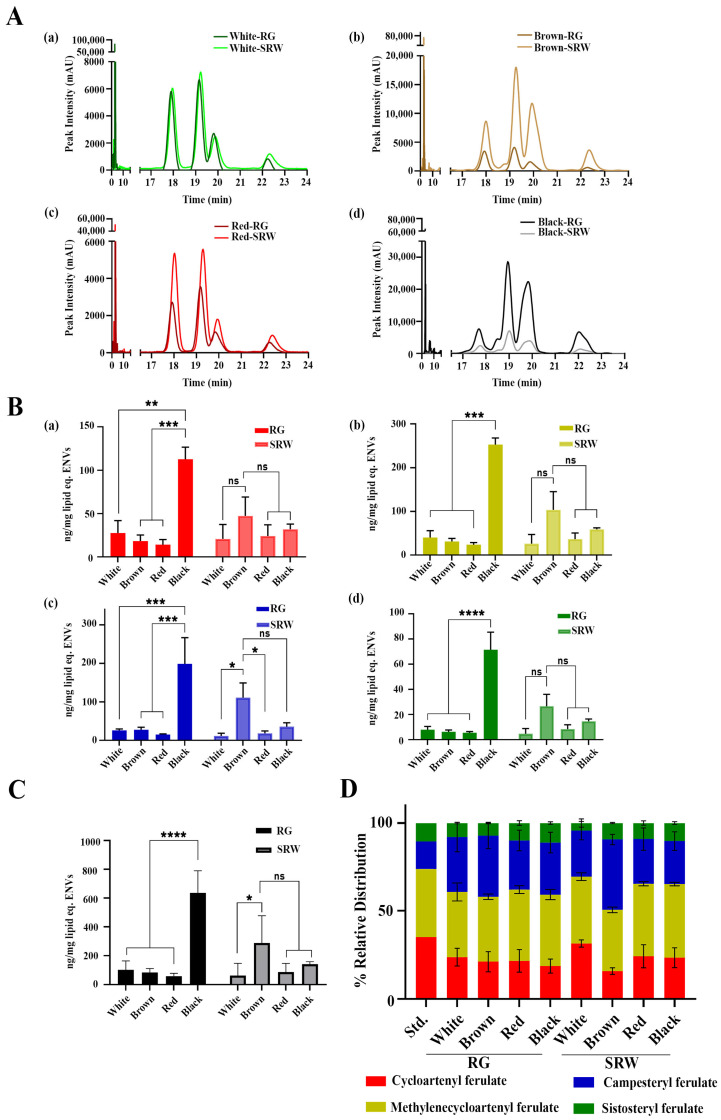
Both RG and SRW PDNVs contain GO. (**A**) HPLC chromatograms of RG and SRW PDNV ethyl acetate extracts showing the presence of all four isomers of GO, namely cycloartenylferulate at 17.9 min, 24-methylenecyclo-artenyl ferulate at 19.3 min, campesteryl ferylate at 19.9 min and β-sitosteryl-ferulate at 22.3 min. (**B**) Total yield of GO isomers in RG and SRW PDNVs per mg of lipid, (**a**) cycloartenylferulate, (**b**) 24-methylenecyclo-artanyl ferulate, (**c**) campesteryl ferylate and (**d**) β-sitosteryl-ferulate. (**C**) Total GO content in RG and SRW PDNVs. (**D**) GO isomer composition in both RG and SRW PDNVs represented as the relative percentage. All figures represent data from three independent PDNV batches isolated from different rice varieties. n.s. means nonsignificant. * *p* < 0.05, ** *p* < 0.01 *** *p* < 0.001, and **** *p* < 0.0001.

**Figure 4 antioxidants-14-00717-f004:**
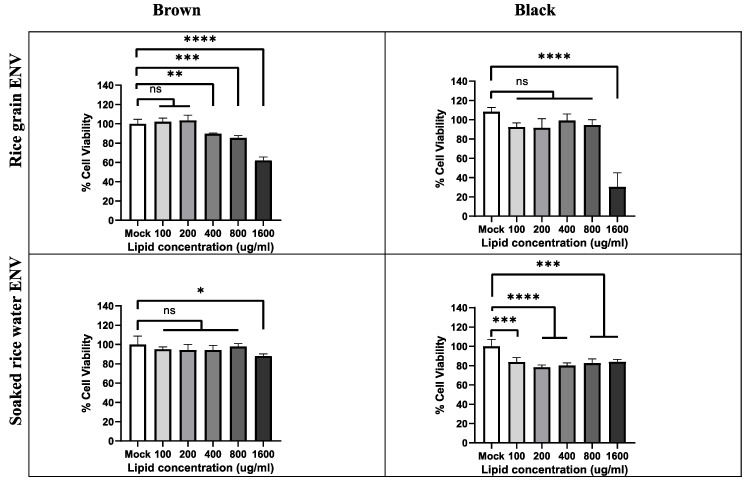
RG and SRW PDNVs are non-toxic to HaCaT cells. HaCaT cells were incubated with either black or brown RGSRW PDNVs or only media (mock) at indicated concentrations for 24 h and relative cell viability was measured by MTT assay as indicated in Materials and Methods. All figures represent data from an average of three independent PDNVs batches; n.s. means nonsignificant. * *p* < 0.05, ** *p* < 0.01, *** *p* < 0.001 and **** *p* < 0.0001.

**Figure 5 antioxidants-14-00717-f005:**
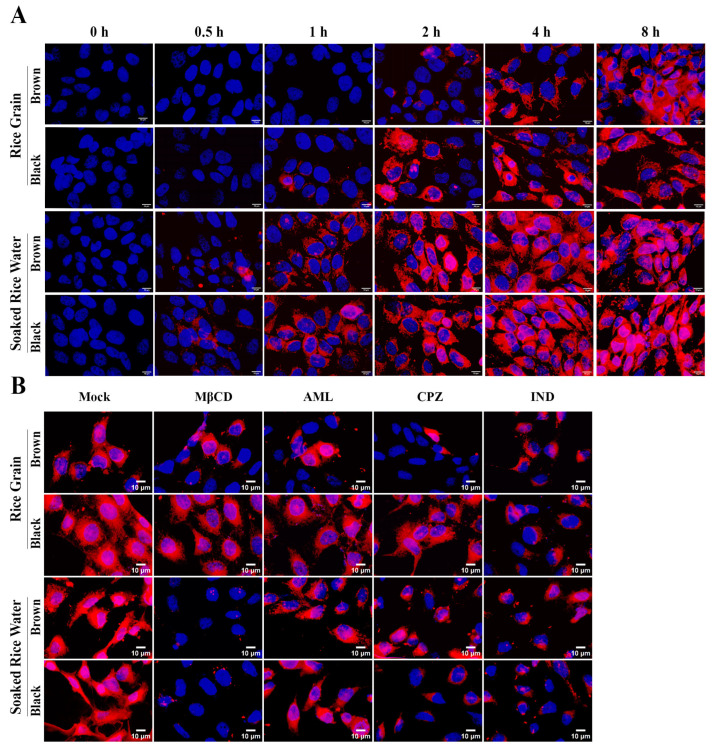
SRW PDNVs display distinct uptake kinetics compared to RG PDNVs. (**A**) HaCaT cells were incubated with 160 μg/mL (lipid equivalent) of DIL-labeled brown or black RG and SRW PDNVs. Cells were washed and fixed at indicated time points, counterstained with DAPI and observed under fluorescent microscope. PDNVs taken up are shown in red and nuclei are shown in blue. (**B**) To dissect out the uptake mechanism, cells were pre-incubated with specific inhibitors of endocytic pathways such as MβCD, AML, CPZ and IND, followed by the addition of indicated PDNVs. Mock refers to cells incubated with PDNV alone.

**Figure 6 antioxidants-14-00717-f006:**
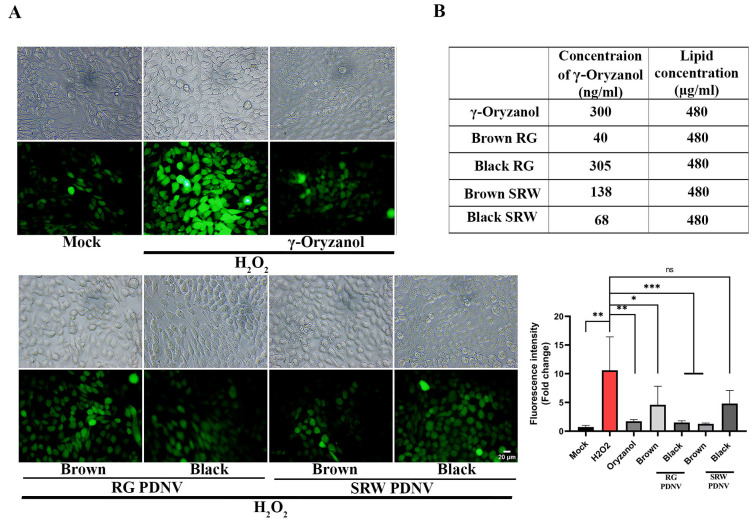
Brown SRW PDNV has potent anti-oxidant activity. (**A**) HaCaT cells were pretreated with 300 ng/mL of γ-Oryzanol or 480 μg/mL of PDNVs for 12 h followed by treatment with H_2_O_2_ to induce oxidative stress. ROS production was determined by treating cells with the ROS-sensitive fluorescent dye, DCFH2-DA, as described in the Materials and Methods section. Representative pictures of ROS production in cells (in green) and corresponding DIC images are shown. Bar graph indicates the relative ROS production measured by quantifying fluorescent intensity taken from multiple random areas within the wells. Data representative of three independent experiments; n.s. means nonsignificant, * *p* < 0.05, ** *p* < 0.01 and *** *p* < 0.001. (**B**) Table depicting the relative oryzanol concentration of PDNVs used in this experiment.

## Data Availability

Data available on request.

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
