# Peer review of "Plant-Derived Nanovesicles from Soaked Rice Water: A Novel and Sustainable Platform for the Delivery of Natural Anti-Oxidant γ-Oryzanol"

_antioxidants, 2025, doi:10.3390/antiox14060717_

Round 1

Reviewer 1 Report

Thank you very much for your interesting research. Some points must be carefully revised. Please, see "detailed comments".

Thank you very much for your interesting research. Some points must be carefully revised:

  1. Line 27. Perhaps it should be indicated that the use of non-eco-friendly solvents is related to “traditional” or “conventional” isolation of GO from rice bran.
  2. Line 50. “Rice bran oil” should be described as raw material.
  3. Lines 58-59. Animal models cannot be used in “clinical” trials.
  4. Line 84. Add reference/s.
  5. Results and discussion. Figure 2. Asterisks or letters should be added to denote statistical differences.
  6. Lines 388-389. Could you please include any examples of these further studies?
  7. Material and methods. Line 401. Were rice grains kernels subjected to any drying process?
  8. Material and methods. Lines 435-455. Is this protein determination measuring total proteins or total soluble proteins?

Reviewer 2 Report

After revision of manuscript: “Plant-Derived Nanovesicles from Soaked Rice Water; A Novel  and Sustainable Platform for the Natural Anti-Oxidant, gamma-Oryzanol Delivery” where author presents isolation of nanovesicles from the subproduct soak rice water and evaluation of antioxidant in cell culture. 

I have some Comments and Suggestions for Authors explained in detailed comments

Lines 7-17. The author’s adscriptions numbers are yellow. Please revise.

Line 34 In the abstract section, you should describe the PEG abbreviation.

Line 58. In Introduction section. It is not correct to mention clinical studies conducted on animal models.

Lines 58-59, 67-71, 128-131. The text is bold. Please revise.

Lines 71-73. It is necessary to add a reference for the affirmation.

In Results & Discussion section. Explain the code S2, S6, and S10 in Figure 1A.

In Results & Discussion section. Explain what is Mock in Figures 4 and 5.

In Results & Discussion section. Please explain the dose dependent decrease in cell viability observed in brown rice grain ENV at concentration of 400, 800 and 1900 ug/mL.

In Results & Discussion section. Figure 6 Line 372. It is necessary to have a space in 300ng/ml. You should add gamma to oryzanol.

In Materials and Methods section. Line 415. Please explain how you can store pellet containing PDNVs

In Materials and Methods section. You should add information about equipment (manufacturer, model, etc.). Example Line 426 (SEM). Line 508 (fluorescence microscope).

In Materials and Methods section. In cell viability and intracellular uptake assays. Explain controls used (negative and positive, if any).

In References section. The references numbers are duplicates. Please revise.
